# RNA Localization and Local Translation in Glia in Neurological and Neurodegenerative Diseases: Lessons from Neurons

**DOI:** 10.3390/cells10030632

**Published:** 2021-03-12

**Authors:** Maite Blanco-Urrejola, Adhara Gaminde-Blasco, María Gamarra, Aida de la Cruz, Elena Vecino, Elena Alberdi, Jimena Baleriola

**Affiliations:** 1Achucarro Basque Center for Neuroscience, 48940 Leioa, Spain; maite.blanco@achucarro.org (M.B.-U.); adhara.gaminde@achucarro.org (A.G.-B.); maria.gamarra@achucarro.org (M.G.); aida.delacruz@achucarro.org (A.d.l.C.); elena.alberdi@ehu.eus (E.A.); 2Departamento de Neurociencias, Universidad del País Vasco (UPV/EHU), 48940 Leioa, Spain; 3Departamento de Biología Celular e Histología, Universidad del País Vasco (UPV/EHU), 48940 Leioa, Spain; elena.vecino@ehu.eus; 4Centro de Investigación en Red de Enfermedades Neurodegenerativas (CIBERNED), 48940 Leioa, Spain; 5IKERBASQUE, Basque Foundation for Science, 48009 Bilbao, Spain

**Keywords:** mRNA transport and localization, local protein synthesis, neurons, neurites, glia, processes, neurological and neurodegenerative diseases

## Abstract

Cell polarity is crucial for almost every cell in our body to establish distinct structural and functional domains. Polarized cells have an asymmetrical morphology and therefore their proteins need to be asymmetrically distributed to support their function. Subcellular protein distribution is typically achieved by localization peptides within the protein sequence. However, protein delivery to distinct cellular compartments can rely, not only on the transport of the protein itself but also on the transport of the mRNA that is then translated at target sites. This phenomenon is known as local protein synthesis. Local protein synthesis relies on the transport of mRNAs to subcellular domains and their translation to proteins at target sites by the also localized translation machinery. Neurons and glia specially depend upon the accurate subcellular distribution of their proteome to fulfil their polarized functions. In this sense, local protein synthesis has revealed itself as a crucial mechanism that regulates proper protein homeostasis in subcellular compartments. Thus, deregulation of mRNA transport and/or of localized translation can lead to neurological and neurodegenerative diseases. Local translation has been more extensively studied in neurons than in glia. In this review article, we will summarize the state-of-the art research on local protein synthesis in neuronal function and dysfunction, and we will discuss the possibility that local translation in glia and deregulation thereof contributes to neurological and neurodegenerative diseases.

## 1. Introduction: What Is Local Protein Synthesis?

Protein synthesis is an essential process for cellular homeostasis. The classical view is that most RNAs in eukaryotic cells are translated within the soma, either in cytosolic free ribosomes or in rough endoplasmic reticulum (RER)-bound ribosomes. Once the proteins are generated and become mature, many of them are targeted to different subcellular compartments where they elicit their function [1]. An accurate targeting of the subcellular proteome is especially relevant in highly polarized, morphologically complex cells where proteins need to be asymmetrically sorted in order to establish distinct structural and functional domains. Subcellular protein distribution is typically achieved by localization signals within the protein sequence (Figure 1A,B). However, protein delivery to distinct cellular compartments can rely, not only on the transport of the protein itself but also on the transport of the mRNA that is then translated at target sites (Figure 1C). Although once considered heretical, mechanisms of localizing RNAs have proven to be highly prevalent and conserved in eukaryotes [2]. Additionally, translation of localized mRNAs, also known as local protein synthesis, is increasingly being recognized as a crucial mechanism that contributes to the physiology of the nervous system (NS) [3].

Proteins need to fulfil three main criteria in order to be considered as locally synthesized in a particular subcellular domain: (1) the mRNAs that encode them and components of the translation machinery (ribosomes, regulatory elements) have to co-localize in the same compartment; (2) de novo protein synthesis should be detected at a subcellular level by techniques such as protein metabolic labeling, puromycilation of nascent peptides or translating ribosome affinity purification (TRAP) (Table 1); (3) protein levels after blocking protein synthesis need to decrease in subcellular domains [4].

mRNAs are delivered to subcellular compartments in association with RNA-binding proteins (RBPs) in a translationally repressed state. Ribonucleoprotein complexes (RNPs) are typically packaged into membraneless supramolecular structures, known as RNA granules, which bind to motor proteins that transport the mRNAs to their final destination at the periphery of the cell. Once the mRNAs reach their target compartment, they are released from the RNP complexes and translated locally into protein by the also localized translation machinery (Figure 1C^i^ and C^ii^) [5]. Hence, to accurately synthesize a protein locally, both RNA transport and translation need to be finely regulated. Malfunctioning of RBPs, deregulation of the translation machinery, changes in the local mRNA repertoire or failure of molecular motors to accurately sort mRNAs to their target compartment can contribute to the disruption of protein homeostasis at a subcellular level. These alterations on the local proteome lead to cellular dysfunction, which in the case of the NS translate into neurological and neurodegenerative diseases [3,6].

Local protein synthesis in the NS has been studied best in neurons, despite neurons not being the only morphologically and functionally complex cells therein. In neurons, local translation occurs both in dendrites and axons. Since the first studies that unambiguously demonstrated the existence of local translation in subneuronal compartments, most interest in the field has focused on the role of locally synthesized proteins in neuronal physiology. However, no evidence existed on the contribution of local protein synthesis to neuronal damage, until almost 20 years ago when activation of intra-axonal protein synthesis in response to nerve injury was reported [7]. Since then, local translation and its defects in neurons have also been linked to amyotrophic lateral sclerosis (ALS) [8,9,10], spinal muscular atrophy (SMA) [11,12,13] or Alzheimer’s disease (AD) [14,15,16,17] to mention but a few NS disorders. Surprisingly, almost no work so far has explicitly reported on the impact of localized translation in glia in neurological or neurodegenerative diseases. In this article we will review the existing evidence of local protein synthesis in neuronal and glial subcellular domains in vertebrates. 

First, we will summarize the current knowledge on local translation in neurons and we will refer to existing data that report mainly on the deregulation of RBPs and the localization or/and localized translation of certain mRNAs in neurological and neurodegenerative diseases. Evidence found in neurons will serve as a canvas to depict the possible contribution of glial localized translation to CNS dysfunction. 

**Table 1 cells-10-00632-t001:** Techniques utilized to measure de novo protein synthesis in neurites.

**Protein metabolic labeling**	**Technique**	**Label**	**Detection**	**References**
	BONCAT/ FUNCAT	Noncanonical aminoacids (azide or alkyne)	Covalent cycloaddition reaction with fluorescently tagged or biotinylated reactive group (alkyne or azide)	[18,19]
	SILAC/ pSILAC	Stable isotope	Mass spectrometry	[20,21]
**Puromycilation of nascent peptides**	SUnSET/ PUNCH-P	tRNA analogue puromycin; biotinylated puromycin	Puromycin immunodetection; biotin-streptavidin conjugation; mass spectrometry following purification of tagged peptides	[22,23,24,25,26]
**Translatomic approach**	TRAP/ Ribo-Seq	Epitope-tagged ribosome	Epitope immunoprecipitation followed by mRNA purification and detection by RNA-Seq	[27,28]

## 2. Neuronal RNA Localization, Local Translation and Nervous System Diseases

### 2.1. Brief Introduction to RNA Localization and Local Translation in Neuronal Processes

Neurons are considered the most morphologically complex cells in the NS. They consist of a cell body or soma from which several processes (dendrites and axons) emerge. Vertebrate neuronal processes can extend from a dozen millimeters (in the case of dendrites) to a meter (in the case of axons) away from the soma [29]. Upon a local signal sensed by neurites, canonical protein synthesis in the cell body followed by protein transport would result in a delayed response from dendrites and axons. Conversely, the presence of localized mRNAs in peripheral neuronal processes allows a rapid reaction of neurites to local stimuli without fully relying on the cell body [5,30]. Local translation in neurons is also involved in the maintenance of the local proteome homeostasis in basal conditions and thus supports basal dendritic and axonal functions [24,31]. 

Local protein synthesis in neurons was first attributed to dendrites, since for many years axons were thought to be devoid of RNAs and ribosomes [30]. Among the first mRNAs detected in dendrites were the ones encoding high molecular weight microtubule-associated protein 2a/b (*Map2a/b*) [32], calcium/calmodulin-dependent protein kinase 2 alpha (*Camk2a*) and the calcium-binding protein neurogranin (*Ng/RC3*) [33,34], the inositol 1,4,5-trisphosphate receptor type 1 (*InsP3R1*) [35], some glutamate receptor and glycine receptor subunits [36,37], and the activity-regulated cytoskeleton-associated protein (*Arc*) [38,39]. mRNAs encoding BDNF, TrkB and ß-Actin are localized to dendrites and /or dendritic spines upon neuronal stimulation [40,41]. More importantly, intra-dendritic synthesis, including the one of β-Actin among other proteins mentioned above, is crucial for nervous system plasticity [42,43].

*Actb* (*ß-actin*) was also the first transcript identified in vertebrate axons in culture [44]. In *Xenopus* embryos, axonal translation of *Actb* contributes to growth cone turning in response to attractive cues and to arbor formation during development [45,46]. Other cytoskeleton- and membrane-associated proteins are locally synthesized upon axon stimulation in rodent neurons: RhoA, Par3, TC10, β catenin and SNAP25 are involved in growth cone behavior [47,48], membrane expansion [49] and synapse formation [50,51]. Local translation is not only required for axon dynamics in response to external cues but also for axon maintenance: axonally synthesized COXIV, LaminB2 and Bcl-W avoid degeneration by regulating mitochondrial function [31,52,53]. Additionally, transcription factors (TFs) can also be locally produced in axons. Two examples thereof include CREB1, which is involved in neuronal survival of dorsal root ganglion neurons [52] and Smad1/5/8, required for axon specification in trigeminal nuclei [53]. As it will be mentioned later, locally synthesized TFs can also be involved in regeneration of peripheral injured nerves [54] and Alzheimer’s-related neurodegeneration [14].

Until recently, local translation in neuronal processes was studied by candidate-based approaches. The transcripts of interest were frequently identified in subneuronal compartments by in situ hybridization and/or quantitative RT-PCR analyses of axonal or neuritic (axonal and dendritic) extracts. Interference strategies and protein synthesis inhibitors were used to determine localized translation and the physiological role of concrete proteins found at subneuronal levels. These studies allowed the characterization of a limited amount of locally synthesized proteins in dendrites and axons. However, the development of massive RNA sequencing techniques has provided a much broader view on the local neuronal transcriptome in different experimental setups. More importantly, recent estimates indicate that ~50% of the local neuronal proteome is generated through translation of localized transcripts [28]. Thus, local protein synthesis in neurons is more extended than it was previously thought. 

Most work on local translation in neurons has focused on the role of locally produced proteins in NS physiology. However, much less has been reported on the role of local protein synthesis in pathologies. In 2001, Zheng and colleagues showed the ability of injured peripheral axons to synthesize proteins when allowed to regrow in vitro. In addition, inhibiting local translation induced the collapse of new growth cones generated in culture [7]. These results strongly suggested that axons could synthesize proteins in peripheral neuropathies, including nerve injury, and that local translation could be involved in axon regeneration and maintenance of lesioned axons. From then on, other studies supported the knowledge that local protein synthesis was affected in damaged peripheral nervous system (PNS) neurons. Finally, as we will review, this knowledge was more recently extended to the CNS as well [6].

### 2.2. Neuronal Local Translation in NS Dysfunction

#### 2.2.1. In Traumatic Nerve Injury

Upon nerve dissection, the proximal portion of the axon forms a growth cone-like structure known as the nerve bulb. Similar to developing axons, severed axons respond to cues in their environment and attempt to grow. mRNAs and components of the translation machinery are recruited to nerve bulbs [6]. Among the proteins which are axonally synthesized after sciatic nerve crush, the TFs STAT3 and PPARγ have been identified. They are then retrogradely transported to neuronal cell bodies where they trigger specific transcriptional programs required for regeneration of injured nerves [54,55]. One of the complexes necessary for the retrograde transport of locally synthesized proteins is importin α/β and importin β is synthesized in lesioned axons too [56]. On the other hand, the master regulator of protein synthesis mTOR is also translated in axons after sciatic nerve injury. A reduction in local mTOR protein levels leads to an overall decrease in protein synthesis in axons. As a result, survival of lesioned neurons becomes compromised [25]. 

In the case of CNS, the redistribution of RBPs to different subneuronal compartments upon damage has been reported. The RNA-binding protein with multiple splicing (RBPMS), which in retinal ganglion cells is expressed exclusively in the soma, becomes sorted to dendrites and axons upon hypoxia and axotomy, respectively [57]. These results indicate that RNA localization to axons is required to adjust the local proteome to the demands of damaged axons also in the CNS. 

Altogether these data suggest that increasing local translation in axons has an important implication in the regeneration of lesioned nerves.

#### 2.2.2. In Motor Neuron Diseases

Whilst changes in the levels of newly synthesized proteins in axons in pathological conditions was first reported in lesioned peripheral nerves [7], recent studies suggest that local translation can also be altered in motor neuron diseases (MNDs). Amyotrophic lateral sclerosis (ALS), also called classical MND, is a disorder whose main feature is the progressive degeneration of upper and lower motor neurons. Among ALS-linked proteins, TDP-43 and FUS can be found in subneuronal compartments. TDP-43 is a RBP that appears aggregated in 97% of ALS patients as well as in many cases of frontotemporal dementia (FTD) [58]. TDP-43 plays a key role in pre-mRNA splicing, mRNA transport and translation in neurites. Studies performed both in *Drosophila* and human samples have identified *Futsch* mRNA (*Map1b* mRNA in mammals) as a TDP-43 target. Interestingly, TDP-43 overexpression leads to a significant reduction of *Futsch* at mRNA and protein levels in neuromuscular junctions while higher levels are observed in the soma of motor neurons. Similarly, MAP1B increases in motor neuron cell bodies in spinal cords from ALS patients. These observations suggest defects in TDP-43 impair *Futsch/Map1b* mRNA localization to axons and its translation [8]. Regarding FUS, it is detected in axons at translation sites and ALS-linked FUS mutations inhibit intra-axonal protein synthesis leading to a stress response with loss of synaptic activity [9]. Interestingly, both TDP-43 and FUS can be found as part of dendritic RNA granules, however the extent to which dendritic localization of both RBPs regulate intra-dendritic translation and contributes to ALS still requires characterization [59]. 

Spinal muscular atrophy (SMA) is a fatal disorder characterized by a progressive degeneration of spinal motor neurons and skeletal muscle atrophy caused by a reduction of survival motor neuron protein (SMN) due to mutations in the human *SMN1* gene. Lack of SMN has been related to mRNA mislocalization in SMN-deficient axons. SMN associates with the RBPs HuD and IMP1 and engages in the assembly of RNP complexes required for mRNA transport to axons. RNP complexes containing SMN recruit mRNAs such as *Actb*, *Nrn1* and *Gap43*, all involved in axon growth and presynaptic function. Thus, SMN loss impairs the formation of RNP complexes and alters axonal mRNA localization and translation, leading to the phenotypic features of SMA [13]. Other mRNAs whose localization is heavily affected by a ~50% loss of SMN are *Anxa2* and *Cox4i2* [11]. Reduced SMN levels also causes an increase in microRNA-183 which leads to reduced levels of axonally synthesized mTOR. As a consequence, mTOR-dependent translation in axons is affected [12]. Translation via mTOR is also regulated by muscle-secreted factors such as C1q/TNF-Related Protein 3 (CTRP3) whose levels are reduced by *SMN1* mutation [60]. 

#### 2.2.3. In Alzheimer’s Disease and Dementia

Alzheimer’s disease (AD) is the leading cause of dementia and it is characterized by the gradual loss of cognitive functions. Despite efforts in developing therapies to stop the progression of this disease, there is still no cure. AD spreads through the brain in a non-random manner indicating propagation along connecting fiber tracts [61], however the molecular mechanisms leading to this spread are poorly understood. What seems clear is that aggregates of extracellular amyloid β peptide (Aβ) and of hyperphosphorylated microtubule associated protein tau (MAPT or Tau) are post-mortem hallmarks of the disease. Thus, Aβ peptides and Tau are considered the main drivers of AD [62,63,64]. In 2014, it was reported that in vitro exposure of axons to Aβ peptides induced local protein synthesis and altered the axonal transcriptome [14]. The mRNAs of 9 out of the ~20 susceptibility genes (ca. 45%) reported in late onset AD were found to be recruited above defined threshold levels to Aβ-treated axons: *App*, *Clu*, *ApoE*, *Sorl1*, *Bin1*, *Picalm*, *Ptk2*, *Celf1* and *Fermt2* [14,60]. Interestingly, local *App* and *Clu* mRNAs were significantly increased upon challenging isolated axons with Aβ compared to vehicle-treated axons [14]. Additionally, the mRNA encoding activating transcription factor 4 (ATF4, previously known as CREB2) is locally translated in response to Aβ oligomers. Axonally synthesized ATF4 is retrogradely transported to the nucleus where it mediates neuronal death by regulating transcription both in vitro and in vivo. Interestingly, *Atf4* mRNA recruitment to axons is itself regulated by local translation of sentinel mRNAs including the intermediate filament protein vimentin (*Vim*) [14,15]. 

*Mapt* is (together with *Actb*) one the earliest examples of an axonally translated mRNA in hippocampal neurons in vitro. Although the presence of *Mapt* mRNA has been reported both in dendrites and axons [17], Tau protein is heavily localized to axons in healthy neurons partially through the localized translation of its transcript [65,66]. In the context of amyloid pathology, Tau is aberrantly translated in dendrites [17]. Increased dendritic translation leads to Tau hyperphosphorylation and the formation of neurofibrillary tangles [16]. Thus, local synthesis of Tau in the inappropriate subcellular compartment could lead to Tau pathology in AD and related disorders.

Besides changes in local mRNA translation, loss of heterogeneous nuclear ribonucleoproteins A/B (hnRNP A/B) has been reported in entorhinal cortices from AD brains. Interestingly, the authors propose that loss of hnRNP A7B might impact on RNA localization in the AD brain based on the observation that hnRNP A2 is part of RNP complexes that deliver mRNAs to the periphery of oligodendrocytes, as will be mentioned later on [67,68]. 

Frontotemporal dementia (FTD) is a group of brain disorders characterized by loss of neurons in the frontal and temporal lobes of the brain. As in AD, FTD patients manifest cognitive impairment but, in some cases, they also develop motor symptoms. Indeed 15% of ALS patients have FTD. Both ALS and FTD are characterized by the accumulation of TDP-43 and FUS aggregates. As stated before, both TDP-43 and FUS are involved in RNA localization and localized translation of dendritic and axonal mRNAs [8,9,65]. Thus, deregulation of protein synthesis caused by malfunctioning of both RBPs could contribute to the development of FTD. Interestingly, in 50% of FTD cases an accumulation of Tau fibrils has also been reported [69]. Since Tau is normally synthesized within axons of healthy neurons but its local translation becomes aberrant in AD, we cannot discard that deregulation of local Tau synthesis is involved in some pathological aspects of FTD as well. 

All these results indicate that unlike peripheral nerve injury, localized translation in AD and related disorders leads to degeneration rather than neuronal maintenance, survival or axon growth. Additionally, evidence mentioned so far suggests that local protein synthesis is more involved in dementias than was previously acknowledged. Thus, in order to develop accurate treatments, mRNA transport and localized translation should not be disregarded.

#### 2.2.4. In Movement Disorders

Huntington’s disease (HD) or Huntington’s chorea is a fatal monogenic neurodegenerative disease characterized by involuntary non-stereotypical movements, as well as behavioral and cognitive impairment. HD is caused by CAG repeat expansions in the Huntingtin (*HTT*) gene resulting in repeated polyQ tracts in the N-terminal region of HTT protein. Mutated HTT is associated with selective toxicity in striatal neurons [70]. The exact role of HTT is not fully understood. Interestingly however, HTT, as well as APP is transported along axons and has been implicated in dendritic RNA delivery [71,72,73,74,75]. Indeed, dendritic HTT co-localizes with the RBPs Ago2 and Staufen, with P-body proteins as well as with the 3’UTR of *Ip3r1*, *Actb*, and *Bdnf* mRNAs. HTT knockout reduces the levels of dendritic *Actb* mRNA, Ago2 protein, and P-bodies. Additionally, HTT suppresses translation of a reporter construct in cortical dendrites. Thus, HTT regulates RNA transport and local translation [73,74,75]. The toxic effect of mutated HTT seems to be driven by its ability to sequester RBPs such as MBNL1 and SRSF6, and therefore mRNA localization is likely impaired in HD. Importantly, aberrant increased protein synthesis is observed in HD mouse models and human brain samples [76,77].

#### 2.2.5. In Fragile X Syndrome, Autism Spectrum Disorders and Intellectual Disabilities

Fragile X syndrome (FXS) is probably the best example of a disease linked to defects in intra-dendritic mRNA translation as reviewed by Swanger and Bassell [59]. FXS is the most frequent monogenic autism spectrum disorder (ASD), accounting for 2% of all cases [78], and is caused by a mutation in the fragile X mental retardation 1 (*FMR1*) gene which causes loss of FMRP protein. FMRP is a RBP that negatively regulates protein synthesis as well as mRNA stability and transport. FMRP associates to dendritic mRNAs encoding well-known synaptic proteins, including *Arc*, *Camk2a*, *Dlg4* or *Map1b*. Therefore, defects in local protein synthesis have been suggested as a key feature underlying FXS, including defects in spine morphology [79]. Interestingly, *Fmrp* mRNA itself is also localized to and translated within dendrites [79]. Moreover, FMRP is expressed in developing and mature axons [80,81,82,83] where its loss alters synaptic connectivity in both *Drosophila* and mouse FXS models [3].

One interesting aspect of the loss of FMRP function is that it impacts mTOR activity, which is not only a key regulator of intra-axonal protein synthesis but also of intra-dendritic translation. Furthermore, mTOR-dependent dendritic translation is suspected to contribute to Rett syndrome, tuberous sclerosis and Down syndrome (DS) [84,85,86]. Interestingly, Down syndrome cell adhesion molecule (DSCAM) is locally produced in dendrites and its localized translation becomes impaired in DS mouse models. Additionally, dendritic BDNF and BDNF-mediated local translation are also increased in DS [59,86,87,88]. All in all, local protein synthesis deregulation has been observed in autistic disorders (including FXS) and intellectual disabilities. 

To sum up, although early work on protein synthesis in subneuronal compartments was mainly performed in neurons under physiological conditions, an increasing body of evidence indicates that mRNA localization and local translation are altered in pathological contexts as summarized in Figure 2. More importantly, locally produced proteins can actively contribute to the pathophysiology of neurological and neurodegenerative diseases as discussed. 

## 3. RNA Localization and Localized Translation in Glia

RNA localization and local translation have been studied far more in neurons than they have in glia. However, neurons are not the only morphologically complex cells in the NS. Glial processes can extend from dozens to some hundred microns from the cell body and can be extremely ramified. It is thus not surprising that in glia too, mRNAs are delivered to distal processes and locally translated into protein in order to maintain the local protein homeostasis and support glial functions. Evidence on local translation in glia was first reported in the early 1980s in oligodendrocytes, in particular in the myelin fraction [89]. 

In astrocytes and radial glia, RNA localization to peripheral process was first addressed over two decades later [90,91]. As in neurons, glial mRNA localization and localized translation have been reported primarily in the healthy CNS, and evidence on the contribution of local protein synthesis or its deregulation to CNS pathology is very sparse. We will, however, review the few existing studies on glial local translation in neurological and neurodegenerative diseases. We will also discuss the possibility that impaired localization of both RBPs and mRNAs to glial processes contribute to CNS dysfunction. 

### 3.1. RNA Localization and Local Translation in Oligodendroglia

Oligodendrocytes enwrap the axons of the CNS with multiple layers of myelin membrane, which increase the nerve conduction efficiency and speed. In addition, these cells provide axons with trophic support, and therefore maintain axonal health and proper cognitive function. Oligodendrocytes are a very stable cell population with the turnover of only 1 in 300 annually. Nonetheless, the myelin is highly dynamic with one cell producing 10^5^ myelin protein molecules per minute and making more than 100 myelin sheaths per cell. This huge biochemical flow is coordinated to ensure the tight temporal and spatial control of the different components of myelin [92,93]. 

In oligodendrocytes, many mRNAs have been reported as being translated under local regulation at the myelin sheath and in distal processes. The myelin basic protein mRNA (*Mbp*) was the first transcript detected in peripheral oligodendroglial processes [89]. Some years later, the mRNAs encoding carbonic anhydrase II (CAII) [94], MAPT/Tau [95], myelin-associated oligodendrocytic basic protein (MOBP) [96] and amyloid precursor protein (APP) [97] were also detected in peripheral domains. Interestingly, *Mbp* and *Mobp* mRNAs are also heavily enriched in myelin, as well as transcripts encoding ferritin 1 (FTH1) and pleckstrin (PLEKHB1) [98]. Oligodendrocytes express an array of RBPs which likely regulate RNA localization and localized translation in order to meet protein requirements in each myelin sheath.

From all localized mRNAs in oligodendroglia, *Mbp* is the most studied due to its importance in proper formation of CNS myelin. *Mbp* mRNA is sorted to the periphery of oligodendrocytes within RNA granules in a translationally inactive state. The sorting mechanism involves the recognition of cis-acting A2 response element (A2RE) in the *Mbp* mRNA by the transacting factor hnRNP A2/B1 [67,68]. Other RBPs including hnRNP E1, hnRNP H/F, hnRNP K and QKI also take part in this process [99,100,101]. Interestingly, the mRNA encoding RBP QKI is one of the most abundant myelin transcripts [98]. Once packaged in RNA granules *Mbp* mRNA is transported along microtubules to peripheral processes and the myelin fraction [102].

Given that many myelin components are transported in specialized RNA granules where RBPs and the microtubule network play critical roles impaired RNA processing, transport failure and/or aggregation of RBPs could affect local translation and even lead to ectopic expression of myelin proteins. Although very little evidence has been published in this regard, we will list below possible links between impaired RNA localization and/or local translation in oligodendrocytes and neurological /neurodegenerative disorders.

#### 3.1.1. In Motor Neuron Diseases

As stated before, 97% of all ALS cases are characterized by the aggregation of the RBP TDP-43 [58]. Interestingly, the vast majority of glial TDP-43 inclusions are found in oligodendrocytes, which are in fact severely affected in this disease [103]. Individuals suffering from ALS, also show a reduction of MBP in both the motor cortex and the ventral spinal cord [104]. Thus, ALS seems not to be a neurodegenerative disease exclusively affecting motor neurons. hnRNP A2/B1 is a known binding partner of TDP-43, and similar to TDP-43, hnRNP A2/B1 has intrinsically disordered aggregation-prone domains which makes it susceptible to fibrillation and incorrect folding. In oligodendrocytes, TDP-43 is physiologically localized in the nucleus, where it binds hnRNP A2/B1. At the same time, hnRNP A2/B1 binds *Mbp* and regulates its distribution towards oligodendroglial processes and the myelin compartments. Therefore, it is feasible that disrupted interaction between TDP-43 and hnRNP A2/B1, due to their pathological aggregation, impairs *Mbp* mRNA transport and its localized translation leading to defects in sheath formation [105,106]. In line with this possibility is the fact that oligodendrocyte-specific deletion of TDP-43 results in defective myelination in mice. While in these mice *Mbp* mRNA remains unchanged, protein levels are decreased, suggesting that MBP protein deficits are caused by deregulation of local translation [107]. 

About 25–40 % of all familial cases and 8 % of sporadic cases of ALS are caused by a hexanucleotide repeat in the chromosome 9 open reading frame 72 gene (*C9orf72*). C9orf72 protein is known to be involved in mRNA metabolism including mRNA translocation between the nucleus and the cytoplasm. Importantly, C9orf72 also binds hnRNP A2/B1 in oligodendrocytes. It is still unclear whether the toxic effects of mutated C9orf72 reported in *C9orf72*-mediated ALS are elicited by a loss of function of the wildtype protein or a gain of function of the mutant one. In any case, impaired mRNA metabolism and sequestering of RBPs have been suggested as pathological mechanisms leading to ALS [105,108]. 

It is worth noting that in ALS patients, as well as in ALS mouse models, myelin pathology precedes axonal degeneration, which suggests that oligodendroglial dysfunction should be targeted earlier than motor neuron pathology [103,104]. From this point of view, understanding the extent to which deregulation of local *Mbp* translation contributes to functional alterations in oligodendrocytes could be crucial to understanding the early stages of ALS. 

#### 3.1.2. In Demyelination:

Similar to C9orf72, the RBP QKI also regulates the shuffle of *Mbp* mRNA from the nucleus to the cytoplasm and its localization to myelin [109,110]. A spontaneous mutation affecting the promoter region of the *QKI* gene in *quaking viable* mice (*qk^v^*) heavily reduces the expression of QKI and decreases the levels of *Mbp* mRNA levels in myelin fractions [109]. Interestingly, these mice present demyelination in different regions on the CNS which has been partially attributed to defects in *Mbp* mRNA localization and the consequent defects in localized translation [111]. Moreover, recent work by Lavon and colleagues has reported that QKI is impaired in the brain of a mouse model of experimental autoimmune encephalomyelitis (EAE) as well as in the blood of patients with neuromyelitis-optica and multiple sclerosis (MS). These observations open the possibility that a dysregulation in the *Mbp* mRNA metabolism due QKI dysfunction could impair local MBP synthesis contributing to demyelinating diseases.

#### 3.1.3. In Alzheimer’s Disease and Dementias

AD has been traditionally considered as a gray matter disease, but during the last decade neuroimaging techniques have revealed micro- and macro-structural changes in white matter (WM), suggesting that, in addition to the neuronal loss, WM degeneration and demyelination are important pathophysiological features in AD, as reviewed by Nasrabady and colleagues in 2018 [112]. More recently, Aβ-induced *Mbp* mRNA local translation was reported in oligodendrocytic processes, establishing a direct link between accumulation of Aβ peptides and aberrant local protein synthesis in oligodendroglia. Local MBP production in distal processes is regulated by Fyn-mediated signaling pathways promoting oligodendrocyte differentiation [113]. Consistent with these observations, the AD APP/PS1 mouse model shows an upregulation of MBP in the hippocampus and increased myelin thickness [114,115]. Taken together, these results suggest that local translation of *Mbp* mRNA could be disrupted in AD, altering myelin morphology and oligodendrocyte development.

Globular glial tauopathy (GGT) is a rare 4R Tau pathology with abnormal accumulation of phospho-Tau in neurons as well as in astrocytes and oligodendrocytes. From a clinical viewpoint GGT shares features with frontotemporal lobal degeneration including FTD. However, motor manifestations have also been reported [116]. Because tauopathy is considered the main driver of the disease we decided to include GGT in this paragraph. Lately, it has been demonstrated that oligodendrocytes are principal targets and dysfunctional players in the pathogenesis of GGT. In GGT cases, a decreased expression of the myelin related proteins MBP, PLP1, CNP, MAG, MAL, MOG, and MOBP has been described indicating that Tau deposits have consequences in oligodendrocyte function and transport of myelin components leading to defective myelin synthesis [117]. Interestingly, the mRNAs encoding all the myelin proteins mentioned are enriched in myelin extracts compared to cerebellar extracts in adult mice suggesting their localized translation [98]. Tau plays a key role in myelination providing the track for the intracellular translocation of myelin products and controlling cell morphology. Loss of Tau in oligodendrocytes disturbs microtubule stability, impairing process outgrowth and intracellular transport [118]. Tau could thus contribute to mRNA localization in oligodendroglia. Interestingly, Tau hyperphosphorylation has been linked to aberrant local translation, at least in neurons [16,17]. Thus, this event could also lead to phospho-Tau accumulation in oligodendrocytes. It is therefore safe to speculate that deregulation of mRNA transport and localized translation in oligodendroglia is more widespread in GGT than was previously acknowledged. 

Although defects in oligodendroglial local translation in FTD have not been suggested so far, as in GGT, Tau accumulation has been reported in 50% of FTD cases [69]. Additionally, the hexanucleotide repeat mutation in *C9orf72* is the most common genetic cause of FTD and is also involved in the development of some AD, HD and PD cases [108]. Overall, these data strongly support the idea that local protein synthesis in oligodendrocytes plays a crucial role in the context of neurodegenerative diseases and thereby, should be studied more in detail.

#### 3.1.4. In Fragile X Syndrome

As previously mentioned, FXS is the most common inheritable form of ASD caused by loss of FMRP [78]. The RBP FMRP selectively binds 4% of brain mRNAs, including *Mbp*, to regulate their transport, translation, and stability [119]. While WM abnormalities have been established in FXS, the effects of FMRP loss in oligodendrocytes and on myelin production are still partially unknown. The specific function of FMRP in mRNA regulation in oligodendroglia is contradictory. Some in vitro and in vivo studies have shown no effect of FMRP loss on *Mbp* translation or myelin components [120], whereas another study has suggested a inhibitory role for FMRP in *Mbp* translation in vitro [121]. Conversely, delayed myelination has been observed in a mouse model of FXS which is consistent with the reported role of FMRP in promoting myelin sheath growth through the local regulation of MBP synthesis in the CNS of *Xenopus* embryos [122,123]. The latter in vivo studies suggest that altered *Mbp* local translation could contribute to FXS. However, in view of current published work, further investigations are needed to clarify the role of FMRP in oligodendroglial local translation in this disease. 

Altogether, the data (summarized in Figure 3) strongly suggest that deregulated local translation in oligodendrocytes plays an important role in neurological and neurodegenerative diseases. Areas of future research must seek to understand the molecular mechanisms underlying local translation in order to develop drugs that selectively control NS disorders involving oligodendroglia dysfunction. 

### 3.2. RNA Localization and Local Translation in Astroglia

Astrocytes are the most abundant type of glia in the mammalian brain. They provide neurons with metabolic, structural and trophic support, they participate in synapse and axon formation and function, and they regulate the cerebral flow by maintaining the blood–brain barrier [124,125]. As neurons, astrocytes are extremely polarized cells with ramifications extending ca. 50 microns away from the soma in rodents and up to 1 millimeter in humans. Astrocytic processes can be classified into branches, branchlets, leaflets and endfeet. Leaflets are thin long processes able to regulate synaptic function and are also known as perisynaptic astrocytic processes (PAPs), whereas endfeet contact the blood vessels, they control blood–brain barrier (BBB) integrity and are also known as perivascular astrocytic processes (PvAPs) [125,126]. Astrocytic processes can be as long as some neurites and it is thus not surprising that they contain RNAs and have the capacity to produce proteins locally. Indeed, early high-throughput analyses of astroglial protrusions revealed that primary mouse astrocytic processes contain ~2200 RNAs at relative levels similar to somatic RNAs or higher, suggesting their enrichment in the periphery of cells [127]. More recently, Sakers and colleagues measured de novo protein synthesis in cortical astrocytes and observed that 73% of translation occurs more than 9 microns away from the nucleus and does not decay in the periphery [111]. Interestingly, years before these studies were performed, mRNA isoforms encoding the glutamate transporter 1 (GLT1) were analyzed in the rat brain by in situ hybridization. *Glt1a* mRNA was detected in abundant levels in astrocytic processes while *Glt1b* mRNA was mainly restricted to the soma [90]. Similarly, *Gfap* isoform α is mainly detected in astroglial protrusions while *Gfapδ* shows mainly somatic localization [128]. Altogether, these results suggest that differential mRNA distribution and localized translation are relevant for functional asymmetry in astrocytes. 

PAPs can contact up to 2 million synapses in humans and thus play a major role in synaptic function [129]. Consistent with this function, a variety of ribosome-bound mRNAs involved in glutamate and GABA metabolism (e.g., *Slc1a2*, *Slc1a3*, *Glul*) as well as in synaptogenesis and pruning (e.g., *Mertk*, *Sparc*, *Thbs4*) were identified in PAPs by Sakers and colleagues. Components of the cytoskeleton and proteins involved in fatty acid metabolism are also locally produced in PAPs according to this same study [111]. A more recent study additionally detected translating mRNAs that encode ribosomal proteins and translation regulators as being enriched in PAPs consistent with results obtained in subneuronal compartments. Interestingly, this study performed by Mazaré and colleagues also described how the repertoire of PAP-translated transcripts were regulated by fear conditioning [130].

Similarly, PvAPs or endfeet, which interact with blood vessels and control BBB integrity and cerebral blood flow, contain mRNAs and are competent for local translation. mRNAs which encode proteins involved in BBB immune quiescence (e.g., *Gja1*), BBB integrity (e.g., *Agt*) and perivascular homeostasis (e.g., *Aqp4*, *Kir4.1*, *Hepacam* and *Mlc1*) are enriched in endfeet [131]. Thus, mRNA localization to PvAPs likely contributes to the regulation of brain vascular physiology.

PAPs and PvAPs share the vast majority of their actively translating mRNAs according to work published in 2020 by Mazaré and colleagues [130]. However, some mRNAs are enriched in PAPs compared to PvAPs including *Ezr*, *Rplp1* and *Fth1* [130]. These results suggest that not only a distinct distribution of transcripts between cell bodies and astrocytic processes contributes to the functional polarity of astroglia, but that processes that support different functions in the brain differ in specific mRNA subsets.

Deregulation of the interaction between astrocytes and neurons and astrocytes and the vascular system leads to the development and progression of many neurological and neurodegenerative diseases as previously reviewed [132,133]. Additionally, the complex morphology of astrocytes is altered in brain disorders [126]. Given that astroglial polarity seemingly relies on mRNA localization and local translation as mentioned, it is expected that locally translated transcripts and/or an aberrant local protein synthesis in astroglial processes are linked to CNS dysfunction as we will review below.

#### 3.2.1. In Motor Neuron Diseases

TDP-43 is a binding protein for the glutamate transporter *Glt1* (*Eaat2*) mRNA and TDP-43 aggregates are associated with astroglial GLT1/EAAT2 loss in ALS mouse models and ALS patients [134,135]. Although TDP-43 aggregation occurs in most ALS cases, in those rare occasions in which its accumulation is absent, patients carry *SOD1* and/or *FUS* mutations. Interestingly, mouse models with ALS-associated *SOD-1* mutations also show reduced levels of GLT1/EAAT2 in astrocytes. Although there is no evidence that decreased astrocytic GLT1/EEAT2 levels are causative of ALS, it seems clear that GLT1 is impaired in this disease [124]. Whether different isoforms of *Glt1* mRNA are selectively affected in ALS suggesting impaired mRNA localization to astrocytic processes and localized translation still requires investigation. 

Additionally, a great number of mRNAs localized to PAPs contain QKI response elements (QREs) [111]. As mentioned before, QKI dysregulation has been involved in demyelination but its role in other diseases such as ALS has not been discarded [136].

#### 3.2.2. In Alzheimer’s Disease and Dementia

Glial fibrillary acidic protein (GFAP) is the most abundant protein expressed in astrocytes. GFAP plays relevant roles in myelination, white matter vascularization and in keeping BBB integrity [137]. Interestingly, immunohistochemical analyses have shown increased expression of both GFAPα and GFAPδ in reactive astrocytes near amyloid plaques in postmortem AD brains, suggesting a role for both proteins in disease progression [138]. *Gfapα* and *δ* mRNAs are distinctly localized in rodent astrocytes: *Gfapα* is restricted to processes while *Gfapδ* is found in the soma. However, in the APPswe/PS1dE9 AD mouse model, the distribution pattern of both mRNAs varies [139]. Moreover, *Gfap* is regulated by the RBP QKI which is upregulated in sporadic AD cases [140]. Thus, *Gfap* mRNA transport and its localized translation are likely disrupted in AD.

Additionally, some of the ribosome-bound mRNAs known to be enriched in PAPs [111,130] are also localized to amyloid-treated axons [14]. Intriguingly, 24.59% of common transcripts (from a total of 130) categorized in significantly represented GO terms (biological process; false discovery rate (FDR) < 0.05) are involved in translation (Figure 4A). These observations strengthen the belief that components of the translation machinery and translation regulators are themselves locally produced. If we narrow down the analysis to those mRNAs significantly changed in amyloid- vs. non-amyloid-treated axons, 21 mRNAs can be found in both the axonal transcriptome and the PAP translatomes (Figure 4B), one of them being *Gfap* and 7 involved in aging (*Lrp1*, *Timp3*, *Aldoc*, *Clu*, *Eef2*, *Fads1* and *Vim*). *Gfap* mRNA is also bound to ribosomes in PvAPs [131]. Other transcripts identified in a PvAP translatome or “endfeetome” are also among the significantly changed transcripts in amyloid-treated axons (*Aqp4*, *Gpm6b*, *Gja1* and *Mlc1*) [14,131]. The coincidence between these datasets is highly suggestive of the presence of a subset of amyloid-responsive mRNAs in astrocytic processes that could contribute to AD. 

PAPs are also enriched in memory-related mRNAs. It would thus not be surprising that their dysregulation could contribute to memory impairment and dementia. Although this point should be confirmed, *Fth1* and *Ftl1* are linked to superficial siderosis and brain iron accumulation, both leading to neurodegeneration and dementia [141], and are bound to PAP ribosomes. Interestingly, both transcripts are reduced upon fear conditioning in mice [130]. 

#### 3.2.3. In Movement Disorders

HTT inclusions are found in striatal astrocytes in a mouse model of HD. Accumulation of HTT in astrocytes coincides with deficits in the inward-rectifying potassium channel Kir4.1 leading to extracellular potassium accumulation in the brain and enhanced neuronal excitability in striatal medium-sized spiny neurons (MSN). Interestingly, overexpression of Kir4.1 in astrocytes partially recovered the HD phenotype in these mice and rescued a number of MSN in the onset of neurodegeneration [142]. Whether *Kcnj10* mRNA (which encodes Kir4.1 protein) localization is altered in HD has not been addressed but given its presence in PvAPs [131], it would be interesting to determine if altered Kir4.1 synthesis in astroglial processes contributes to the manifestation of HD symptoms and the progression of the disease.

#### 3.2.4. In Fragile X Syndrome

*Fmr1* knockout mice which model the loss of FMRP function in FXS show a significant protein synthesis-dependent reduction in the glutamate transporter GLT1/EEAT2 and in glutamate uptake. The involvement of astroglial GLT1 to this phenotype was reported by Higashimori and colleagues in 2016. Importantly, loss of astroglial FMRP contributes to GLT1 dysregulation, impaired glutamate uptake, cortical synaptic deficits and other FXS phenotypes [143,144]. These results point towards the possibility that *Glt1* mRNA localization and translation are altered in astrocytic processes, however addressing whether the loss of the process-specific *Glt1* mRNA isoform (*Glt1a*) [90] contributes to FXS would lead to more conclusive results.

#### 3.2.5. In Other Neurological Disorders

*Aqp4* and *Gja1* (which encodes the hemichannel protein Cx43) deregulation might not only be involved in AD as previously mentioned. Both transcripts are locally translated in PvAPs and these astroglial peripheral structures play relevant roles in BBB maintenance and immunoregulation. In some neurological conditions the BBB integrity becomes compromised and polarized expression of Aqp4 and Cx43 is lost suggesting that their localized translation in PvAPs might be affected [145]. For example, in the mouse model of middle cerebral artery occlusion (MCAO), Aqp4 is increased during edema and ischemia and its polarization is lost, whereas Cx43 increases in peri-infarct area several days after MCAO. Both proteins are thus potential therapeutic targets for stroke [146,147,148]. In MS BBB, dysfunction is also evident, and astrocytes are considered active participants in the progression of the disease rather than simple scarring cells [149]. Interestingly, Aqp4 is reduced in endfeet and Cx43 polarity is lost [145]. Finally, loss of perivascular Aqp4 and its mislocalization have been associated to epilepsy in the latent phase preceding seizures [150]. Altogether, these results suggest that mislocalization of mRNAs and defects in localized translation in PvAPs are associated with loss of BBB integrity in some neurological conditions and might contribute to disease progression.

The data reviewed in this paragraph (and summarized in Figure 4C and Table 2) add to a mounting body of evidence indicating the relevance of local protein synthesis in astrocytic processes in proper NS function.

### 3.3. RNA Localization and Local Translation in Radial Glia

Radial glia cells (RGCs) are the neural progenitors of the developing cortex. Their bipolar structure comprises a cell body located close to the ventricular surface (apical) of the neuroepithelium, a basal process and an endfoot that contacts the pial surface (basal) (Figure 5). In contrast to neuroepithelial cells, RGCs express astroglial markers but have both neurogenic and gliogenic potential [151,152,153]. RGCs receive numerous signals from cells located at the basal side of the epithelium including Cajal-Retzius cells, meningeal cells and excitatory upper layer neurons. It is believed that these signals released by the basal niche reach the endfeet which in turn “transfer” them to the cell bodies located near the ventricles and 150–450 µm away from the pial surface [154,155,156].

Hence, the basal process would work as a communication road between the basal and the apical sides of the cell [156,157]. Some mRNAs are locally translated into protein in RGC endfeet as reviewed by Pilaz and Silver in 2017 [158]. Among these transcripts those encoding nuclear proteins such as cyclin D2 (*Ccnd2*) have been found, suggesting that proteins translated locally in endfeet could elicit their function in the nucleus as a means of communication between the pial and the ventricular surfaces of the neuroepithelium [91,156]. Indeed, a similar mechanism involving transcription factors has been reported in neurons in both physiological and pathological contexts [14,52,53,54]. Transcripts for cytoskeleton components and signaling molecules are also present in endfeet [156].

The only characterized RBP able to transport mRNAs to RGC endfeet is FMRP [156]. Given that loss of FMRP is associated to FXS among other diseases, we will discuss below the possibility that dysregulation of endfeet FMRP targets participates in NS dysfunction.

#### 3.3.1. In Fragile X Syndrome, Autism Spectrum Disorders and Intellectual Disabilities

Transcriptome analyses performed so far in RGC endfeet have focused on targets of the RBP FMRP [156]. Pilaz and colleagues performed RIP-Seq and identified 115 potential FMRP targets. Interestingly, 31% of the identified transcripts are involved in neurological diseases, more than 20 of them being encoded by ASD-related genes. Moreover, some pulled-down transcripts are associated with intellectual disabilities. From all identified FMRP-bound mRNAs, six were further validated by in situ hybridization: *Vash1*, *Ptpn11*, *Apc*, *Kif26a*, *Dst* and *Campsap2*. Importantly *Kif26a* mRNA localization to endfeet and transport are impaired in *FMR1* knockout animals. Taken together, these results (summarized in Figure 5) suggest that mRNA localization to and localized translation within endfeet are affected in FXS, other ASDs and likely in intellectual disabilities.

#### 3.3.2. In Other Neurological Disorders

As stated before, *Ccnd2* is locally translated in RGC endfeet and is also an FMRP target [91,156]. Additionally, mutations in the *CCND2* gene have been linked to megalencephaly-polymicrogyria-polydactyly-hydrocephalus syndrome (MPPHS), a rare neurodevelopmental disorder that leads to severe brain malformations [159]. It would be interesting to address whether megalencepahly syndromes are accompanied by dysregulation of local CCND2 production in RGC endfeet during cortical development.

To sum up, although local protein synthesis in RGCs has been by far less studied than it has in other neural cells, the limited existing evidence on this phenomenon (Figure 5 and Table 2) reveal the intriguing possibility that local translation in processes of neural progenitors has important implications in brain development. More interestingly, dysregulation of mRNA localization and localized translation in these cell types could contribute to neurodevelopmental disorders.

## 4. Concluding Remarks

The information gathered herein (summarized in Table 2) regarding localized translation in glial processes indicate that despite this phenomenon not being extensively studied, there is already a decent amount of evidence suggesting that dysregulation of mRNA localization and local translation in glia might significantly contribute to NS dysfunction. It is worth mentioning that the fact that microglia have not been mentioned in this review article relates to the lack of evidence that local protein synthesis occurs in this cell type until fairly recently [160]. Although it is still too early to venture that RNA localization and/or local translation in microglia play significant roles in CNS disease progression, it is foreseen that this cell type will definitely emerge into the picture of localized translation soon and will deserve future mention. This possibility, together with the evidences reviewed in this article, open new and exciting avenues for the search for novel (and localized) therapeutic targets for neurological and neurodegenerative diseases involving RNA localization and localized translation in glia.

## Figures and Tables

**Figure 1 cells-10-00632-f001:**
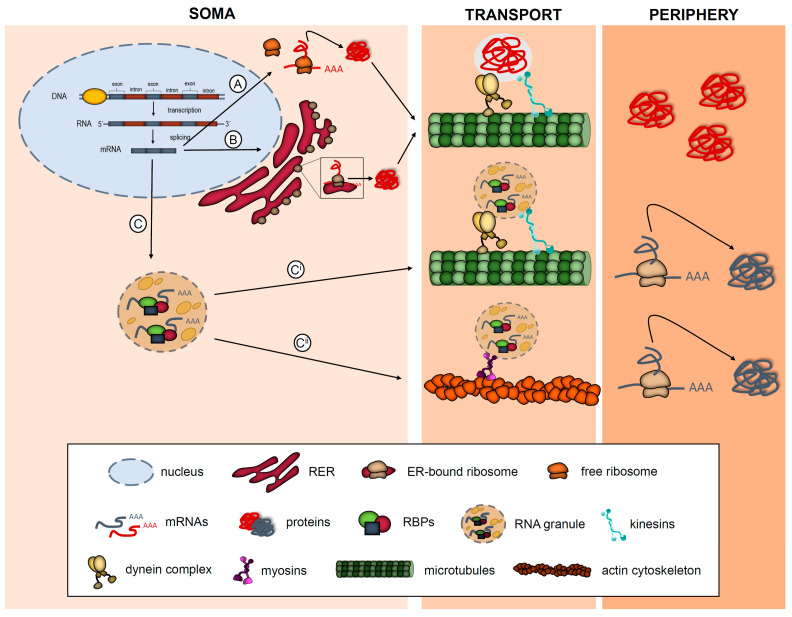
Simplified model of protein and mRNA trafficking in eukaryotic cells. Typically, proteins are thought to be synthesized in the soma by free ribosomes (**A**) or endoplasmic reticulum (ER)-bound ribosomes (**B**) and then transported to different destinations in the cells where they elicit their function. Some mRNAs associate to RNA-binding proteins (RBPs) and are transported in RNA granules (**C**) by microtubules using motor proteins kinesis and dynein (**C^i^**) or by actin using myosin (**C^ii^**). Once they reach the target compartment the mRNAs are translated into protein.

**Figure 2 cells-10-00632-f002:**
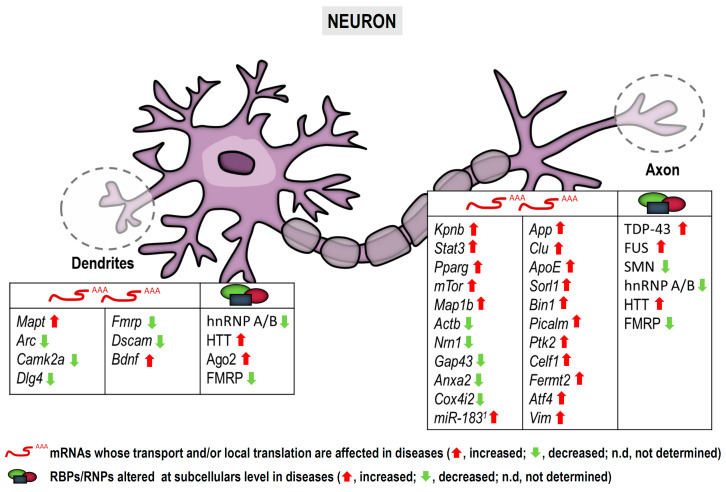
Neurite-localized transcripts and RBPs affected in pathological conditions. Note that some mRNAs and RBPs depicted differ between dendrites and axons. ^1^ Indicates miR-83 which is not an mRNA but a microRNA, yet it is localized to axons and regulates local protein synthesis [12]. Arrows indicate altered levels (red, increase; green, decrease) of mRNAs and/or proteins in diseases.

**Figure 3 cells-10-00632-f003:**
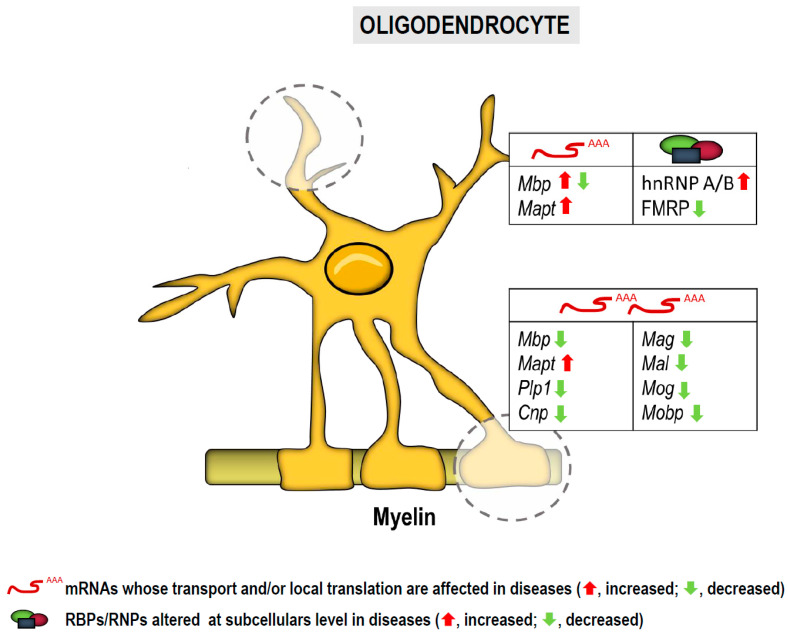
Transcripts and RBPs localized to subcellular compartments in oligodendroglia potentially involved in neurological and neurodegenerative diseases. Local translation occurs both in processes (upper table) and in the myelin sheath (lower table). Arrows indicate altered levels (red, increase; green, decrease) of mRNAs and/or proteins in diseases.

**Figure 4 cells-10-00632-f004:**
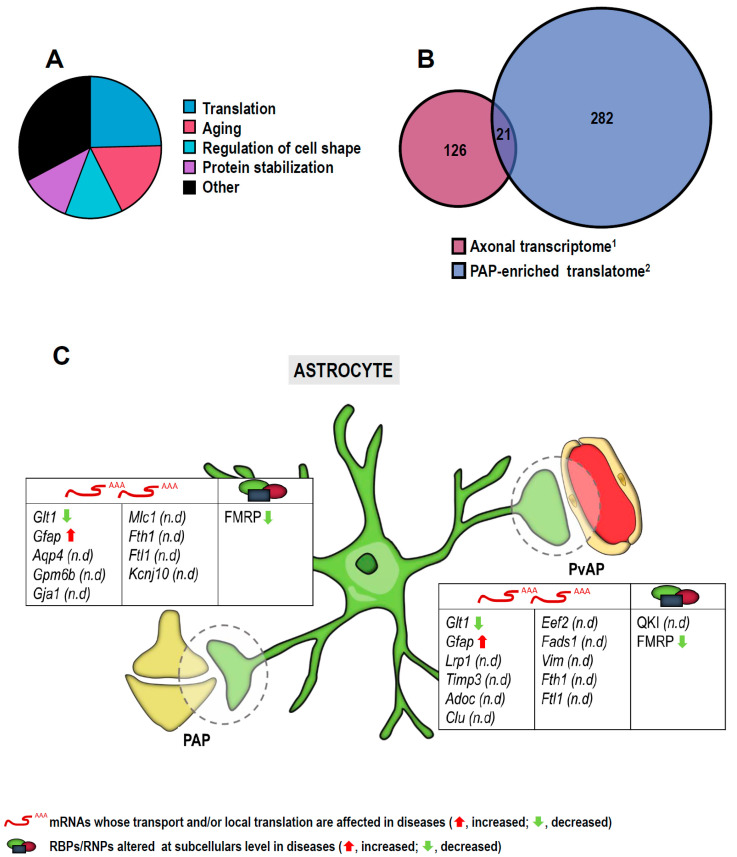
Taking advantage of the detailed analyses performed on the local translatome in astroglial peripheral processes [111,130] and on the axonal transcriptome of Aβ-treated axons [14], a GO term analysis of common localized transcripts found in astrocytes and neurons is shown (FDR < 0.05) (**A**). Focusing on significantly changed transcripts in Aβ-treated axons compared to controls (^1^), 21 one of them are also found in perisynaptic astrocytic process (PAP) translatome datasets (^2^) (**B**). Local translation occurs both in PAPs (upper table) and in perivascular astrocytic processes (PvAPs) (lower table). Arrows indicate altered levels (red, increase; green, decrease) of mRNAs and/or proteins in diseases (**C**). n.d, not described.

**Figure 5 cells-10-00632-f005:**
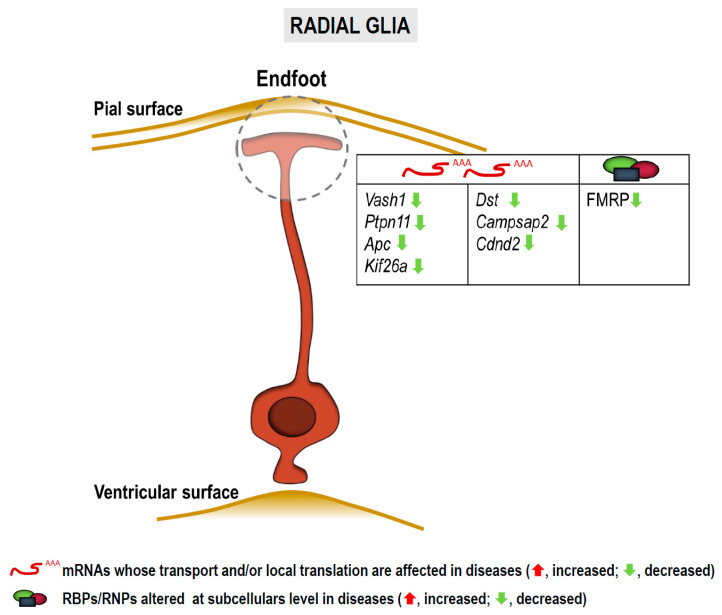
Transcripts and RBPs localized to subcellular compartments in radial glia potentially involved in neurological and neurodegenerative diseases. Arrows indicate altered levels (green, decrease) of mRNAs and/or proteins.

**Table 2 cells-10-00632-t002:** List of RNAs and RBPs whose localization and /or local translation is proven or suggested to be linked to neurological and/or neurodegenerative diseases. AD, Alzheimer’s disease; ALS, Amyotrophic lateral sclerosis; ASD, Autism spectrum disorder; DS, Down syndrome; FTD, Frontotemporal dementia; FXS, Fragile X syndrome; HD, Huntington’s disease; MS, Multiple sclerosis; SMA; Spinal muscular atrophy. Others include stroke, dementia, intellectual disabilities and neurodevelopmental disorders. N, neurons; A, astrocytes; R, radial glia; O, oligodendrites. N^1^ refers to significantly changed transcripts in Aβ-treated axons vs. controls that have not been further referred to in the literature and are thus not included in Figure 2. MS^2^ includes MS and demyelination.

List of mRNAs/microRNAs Localized to Subcellular Compartments in Neurons and Glia and Implicated in Diseases			
Transcript	Disease	Cell Type	References
*Actb*	SMA, HTT	N	[13,73,75]
*Aldoc*	AD	N^1^, A	[14,111,130]
*Anxa2*	SMA	N	[11]
*Apc*	FXS	R	[158]
*ApoE*	AD	N	[6,14]
*App*	AD	N	[6,14]
*Aqp4*	AD, MS, other	N^1^, A	[14,131,145,150]
*Arc*	FXS	N	[79,80]
*Atf4*	AD	N	[14]
*Bdnf*	HTT, DS, other	N	[59,86]
*Bin1*	AD	N	[6,14]
*Camk2a*	FXS	N	[79,80]
*Campsap2*	FXS	R	[156]
*Ccdn2*	FXS, other	R	[156,159]
*Celf1*	AD	N	[6,14]
*Clu*	AD	N, A	[6,14,111,130]
*Cnp*	GGT	O	[117]
*Cox4i2*	SMA	N	[11]
*Dlg4*	FXS	N	[79,80]
*Dscam*	DS	N	[59,87]
*Dst*	FXS	R	[156]
*Eef2*	AD	N^1^, A	[14,111,130]
*Fads1*	AD	N^1^, A	[14,111,130]
*Fermt2*	AD	N	[6,14]
*Fmrp* (*Fmr1*)	FXS	N	[79,80]
*Fth1*	other	A	[141]
*Ftl1*	other	A	[141]
*Gap43*	SMA	N	[13]
*Gfap*	AD	N^1^, A	[6,14,111,130,131,139]
*Gja1*	AD, MS	N^1^, A	[14,131,145]
*Glt1*	ALS, FXS	A	[124,143,144]
*Gpm6b*	AD	N^1^, A	[14,131]
*Kpnb*	TNI	N	[56]
*Kcnj10*	HTT	A	[142]
*Ki26a*	FXS	R	[156]
*Lrp1*	AD	N^1^, A	[14,111,130]
*Mag*	GGT	O	[117]
*Mal*	GGT	O	[117]
*Mapb1*	ALS, FXS	N	[8,79,80]
*Mapt*	AD, GGT	N, O	[16,17,117]
*Mbp*	ALS, MS^2^, AD, GGT, FXS	O	[107,110,111,113,117,122,123]
*miR-183*	SMA	N	[12]
*Mlc1*	AD	N^1^, A	[14,131]
*Mobp*	GGT	O	[117]
*Mog*	GGT	O	[117]
*mTor*	TNI, SMA, ASD, DS	N	[12,25,84,85,86]
*Nrn1*	SMA	N	[13]
*Picalm*	AD	N	[6,14]
*Plp1*	GGT	O	[117]
*Pparg*	TNI	N	[55]
*Ptk2*	AD	N	[6,14]
*Ptpn11*	FXS	R	[156]
*Sorl1*	AD	N	[6,14]
*Stat3*	TNI	N	[54]
*Timp3*	AD	N^1^, A	[14,111,130]
*Vash1*	FXS	R	[156]
*Vim*	AD	N, A	[14,15,111,130]
Ago2	HD	N	[73,74,75]
FMRP	FXS	N, O, A, R	[79,80,122,133,143,144,156]
FUS	FTD	N	[8,9,59]
hnRNP A/B	AD, ALS	N, O	[67,68,105,106,107,161]
HTT	HD	N	[73,74,75]
QKI	MS, ALS, AD	O, A	[110,111,136,140,162]
RBPMS	TNI	N	[57]
SMN	SMA	N	[13]
TDP-43	ALS, FTD	N, A	[8,9,59,124]

## Data Availability

Not applicable.

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
