# Peer review of "RNA Localization and Local Translation in Glia in Neurological and Neurodegenerative Diseases: Lessons from Neurons"

_cells, 2021, doi:10.3390/cells10030632_

Round 1

Reviewer 1 Report

The authors provide a very comprehensive overview of the role of local translation in brain homeostasis and how this can be impacted in disease. Whilst the authors were thorough and provided well cited literature, I question the novelty that this review brings to the field. The review is also way too long.

The title of the review focusses on local translation in glia yet the largest section of the review summarises neuronal local translation? I understand the authors were attempting to use what's known in neurons to extrapolate to the understudied glia; however, the authors didn't actually do this. I suggest significantly reducing the neuronal section, particularly given it has been extensively reviewed elsewhere so there is no novelty to this section of the review.

Some sections of the review don't actually raise any relevant points. For example, the MND and HD sections in the astrocyte part of the review don't really reference RNA trafficking. I can appreciate that it because it remains understudied in this context but then why include it? The radial glia section only contained those diseases that had been studied with minimal extrapolation so I suggest this same approach be applied to all sections.

I'm confused as to the point of Fig 4A&B - they require more explanation of their relevance.

Why haven't microglia been mentioned?

Author Response

We thank the reviewer for her/his imput. We have attached a point-by-point response below.

Reviewer 2 Report

The authors first overviewed defects in mRNA localization and local translation in neurons in several neurological and neurodegenerative diseases. They then cataloged aberrant regulation of mRNA localization and translation in several types of glia in diseases. This is an interesting review, as there are many reviews on local translation in neurons, but far fewer on glia.

1. In Figures 2-5, mRNAs and RBPs are listed, but it is difficult to read how diseases affect them. For example, what are the normal and abnormal states of Mapt mRNA? Mapt is listed in dendrites, and this classification can be read in several ways: 1. Mapt is usually localized to dendrites and its translation is affected by diseases. 2. Mapt is usually localized to axons and, in diseases, mislocalized to dendrites. 3. Mapt is usually localized to dendrites and the localization is affected by diseases. I recommend the authors to clearly show the localization of the mRNAs and RBPs in health and what happens with the diseases.

2. In section 3.1.2, the authors described QKI. In qkv mice, the translocation of Mbp mRNA from the nucleus to the cytoplasm is affected, which does not appear to be related to local translational regulation of MBP by QKI. I felt that this section did not match this review. In addition, in Figure 3, QKI appears to be localized to the myelin sheath or to regulate mRNA localization and local translation there. If there are some studies suggesting them, refer to the papers. Otherwise, listing QKI in Figure 3 may be misleading.

3. The legend in Table 2 does not explain what the cell type indicates.

Author Response

We thank the reviewer for her/his imput. We have atteched a point-by-point response below.

Reviewer 3 Report

This is a nice review of the literature with a particular focus on local translation on glia. The manuscript is up to date, well balanced, and nicely organized and should be of interest to the wide audience of Cells Journal. I have the following suggestions that the authors might consider to make this manuscript even more  valuable.

1. line 34: " ... occurs in the rough endoplasmic reticulum (RER)... ".
Isn't this only true for transmembrane and secreted proteins? Cytosolic proteins are translated by free ribosomes, not associated with ER. I think that this sentence, as well as Figure 1, should be revised.

2. line 78: "..role of locally-synthesized proteins in neuronal physiology. However, no evidence existed on the contribution of local protein synthesis  or its deregulation to neuronal pathology, until almost 20 years ago when activation of intra-axonal protein synthesis in response to nerve injury was reported [7]. "
This reference reports the requirement of local protein synthesis in axon regeneration. Although this is indeed one of the important discoveries, I am not sure whether the inability to regenerate axons after axotomy can be described as "pathology". One could perhaps say that this paper was one of the first studies of local protein synthesis performed with a direct focus on its relavance on the nervous system disorders or damages?

3. Table I.
It would be better to edit the contents of the "Technique" for better readability (e.g. a new line after "/")

4. line 134.
Would "by candidate-based approaches" be a better term for "in a one-to-one basis"?

5. line 158.
could be able to --> could

Author Response

(The authors gave the same response as above.)

Round 2

Reviewer 1 Report

Thank you for your responses.